# Encouraging perspective taking: Using narrative writing to induce empathy for others engaging in negative health behaviors

**Victoria A. Shaffer** [1] *, **Jennifer Bohanek**[1], **Elizabeth S. Focella**[2], **Haley Horstman**[3], **Lise Saffran**[2]

**1** Department of Psychological Sciences, University of Missouri, Columbia, MO, United States of America,
**2** Department of Health Sciences, University of Missouri, Columbia, MO, United States of America,
**3** Department of Communication, University of Missouri, Columbia, MO, United States of America

* shafferv@missouri.edu

**Data Availability Statement:** Data, experimental materials, and R markdown documents detailing analyses are fully available at: https://osf.io/8fhrn/.

## Abstract

Societal expectations of self-care and responsible actions toward others may produce bias against those who engage in perceived self-harming behavior. This is especially true for health professionals, who have dedicated themselves to helping reduce the burden of illness and suffering. Research has shown that writing narratives can increase perspective taking and empathy toward other people, which may engender more positive attitudes. Two studies examined whether creating a fictional narrative about a woman who smokes cigarettes while pregnant could increase positive attitudes toward the woman who smokes and reduce the internal attributions made for her behavior. Across both experiments, the narrative writing intervention increased participants' empathy and perspective taking, evoked more positive attitudes toward a woman who smokes cigarettes while pregnant, and increased external attributions for her behavior. This work supports our hypothesis that narrative writing would be an efficacious intervention promoting attitude change toward patients who engage in unhealthy, and often contentious, behaviors. This work also suggests that narrative writing could be a useful intervention for medical professionals and policy makers leading to more informed policy or treatment recommendations, encouraging empathy for patients, and engendering a stronger consideration of how external forces can play a role in someone's seemingly irresponsible behavior.

## Introduction

Imagine witnessing someone you know who struggles with alcoholism order a drink at a bar. Imagine a parent using drugs in front of their child or a pregnant woman who smokes. Witnessing these behaviors may make observers uncomfortable and lead to negative attitudes toward the individual engaging in them [1]. The development of these negative attitudes may be particularly problematic when they occur in the context of helping professions (e.g. psychology, medicine, social work). Previous research has indicated that health professionals can form negative attitudes towards patients who engage in negative health behaviors, such as drug

**Funding:** The authors received no specific funding for this work.

**Competing interests:** The authors have declared that no competing interests exist.

use and substance misuse [2, 3] and toward those that are obese [4]. To the extent that unhealthy behavior, along with other health-related outcomes, conforms to the social and economic gradient in health—with less educated and less wealthy populations being more at risk—the negative attitudes of providers may be disproportionally attached to marginalized populations, especially in cases where providers have received only superficial training in the social determinants of health [5, 6]. There is great concern that the attitudes and beliefs of providers will influence the practice of health care and contribute to the documented health disparities [7–9]. Therefore, it is imperative to understand both the reason that providers form these negative attitudes toward their patients and methods for preventing or reversing the process.

Research has shown that one effective way to influence health professionals' relationships with their patients and improve the care they provide is by increasing empathy [10]. Empathy can be defined as the ability to identify another's emotional state and feel what the person is feeling. Thus, it includes elements of perspective-taking such as imagining yourself in someone else's situation, acknowledging their point of view, seeing things through their eyes, and trying to understand their emotional state and behavior [11]. Research has shown that a practitioner's ability to empathize is a trait preferred by some patients [12] and that the ability to take the perspective of patients is associated with greater patient satisfaction [13]. Further, physicians who score highly on a measure of empathy have patients with more positive clinical outcomes for diabetes [14]—however, there also exist discordant data [15]—and patient ratings of provider empathy are associated with better health outcomes, including a shorter and less severe experience with the common cold [16, 17].

Although higher levels of physician empathy are related to better outcomes for patients, and patients report empathy as an important aspect of their care, research has indicated that not all physicians are able to empathize with their patients [18, 19], and empathy tends to surprisingly decrease across medical school and residency [20]. Further, even when a physician may show empathy toward one patient, they may not do so consistently across patients [21].

One method of increasing empathy is to share in a common experience. Research has demonstrated that when physicians experience a serious illness and become patients themselves, they develop increased empathy, place a greater emphasis on the patient's preferences, and demonstrate a willingness to work *with* the patient through the decision-making process [22, 23]. Further, reflective writing, which seeks to draw out personal experience with illness, has also been a useful method of inducing this type of empathic personal reflection [24].

In addition to the use of reflective writing on personal experience, a number of interventions have been designed to build empathy in health professionals [10, 20], including the use of creative writing exercises [25–27]. While reflective writing typically asks the writer to reflect on their own personal experiences, the purpose of creative writing is to shift the focus from the writer to an external character, often a patient, allowing the writer to develop an empathic affiliation [28, 29]. For example, medical students who engaged with elderly patients diagnosed with dementia in the context of a creative storytelling task felt more empathy and had more positive attitudes towards these patients, as well as dementia patients more generally [29]. Additionally, Shapiro and colleagues compared essays written by two groups of medical students, those trained in creative writing and those trained in clinical reasoning [30]. Stories written by both groups were analyzed with both thematic coding and the Linguistic Inquiry and Word Count program [31] and demonstrated that students trained in creative writing wrote stories with more emotion, empathy, insight, and religious and spiritual references than the students in the clinical reasoning group.

In addition to increasing empathy, creative writing exercises also promote perspective taking, or taking the perspective of another. There are a number of benefits associated with perspective taking, including: 1) reducing reliance on stereotypes to judge members of an

outgroup [32] and 2) creating a stronger link between the self and the outgroup (termed self-outgroup merging), which can result in more positive attitudes toward the outgroup [33]. Additionally, perspective taking can help debias the fundamental attribution error, which describes our tendency to view the negative behaviors of others as a function of stable, internal attributes or personal traits (e.g., they are a bad person), while we view our own negative behaviors as a reflection of external factors (e.g., I had very little sleep last night). Taking the perspective of an outgroup member results in fewer internal attributions for their situations or behaviors [34] compared to those who do not engage in perspective taking. Further, studies have found that some of the positive effects of perspective-taking can last for hours [35], days [36], weeks [37], and even months [38].

Importantly, increasing empathy and perspective-taking not only shifts our attitudes and attributions towards others but also has the potential to reduce health disparities and change the care that marginalized groups may receive. In a compelling study examining disparities in pain management, Drwecki and colleagues demonstrated that both undergraduate and registered nurse participants exhibited a bias favoring White patients over African-American patients in the (fictional) pain management treatment they would assign when simply told to make their "best, most accurate treatment decisions for each patient" [39]. However, participants assigned to a perspective-taking condition, who were given several additional prompts including "try to imagine how your patient feels about his or her pain and how this pain is affecting his or her life", did not exhibit a pain treatment bias against the African-American patients. Instead, participants in the perspective taking condition assigned similar pain treatments to both White and African-American patients.

Because a lack of empathy, limited perspective taking, and biased attribution of causes to behaviors is likely to impact the care a patient receives and their subsequent health outcomes, it is important to develop strategies for the creation of more empathic, and less biased, health care providers. To date, there has been no research targeting these three psychological constructs (empathy, perspective taking, and the fundamental attribution error) in the context of health behavior. To address this gap in the literature, we propose the use of a creative narrative writing intervention designed to encourage writers to reflect on the social determinants of health behaviors [40, 41]. The target of this writing exercise is a fictional character that engages in a negative, and controversial, health behavior: smoking while pregnant. Because of the detrimental effects on fetal development [42], this is a behavior that may not only harm a woman herself, but potentially her unborn baby.

Drawing upon the narrative and creative writing literatures, we hypothesized that writing a fictional narrative about a person who engages in an unhealthy behavior would increase empathy and perspective taking (H1) and make attitudes towards the person engaging in a negative health behavior more positive (H2). Further, we hypothesized that creative narrative writing would debias the fundamental attribution error (H3), whereby participants would be less likely to make internal attributions about the person's behavior and more likely to recognize the influence of external factors.

To test these three hypotheses, we conducted two studies in which participants constructed a fictional narrative about a woman who smokes cigarettes while pregnant and examined its effect on participants' empathy/perspective taking, attitudes towards the woman who smokes while pregnant, and internal vs. external blame. The purpose of this research was to provide "proof of concept"—that this creative narrative writing intervention can produce the hypothesized change in these psychological constructs. Therefore, the samples of our two studies are comprised of undergraduate college students. However, if this creative narrative writing exercise is efficacious, it could form the basis of an intervention for healthcare providers that is

designed to increase empathy, promote perspective taking, and debias the fundamental attribution error.

Study 1 used a within-subjects design, measuring participant responses before and after the narrative writing intervention. Study 2 was designed to replicate the findings of Study 1 as well as to test the hypothesis that the observed changes from before to after the narrative writing intervention were the result of the narrative intervention itself and not an artifact of repeated measurement. To do so, Study 2 employed a mixed design where we compared pre-post responses on empathy/perspective taking, attitudes, and internal vs. external attributions for both the narrative writing condition that employed the narrative writing intervention from Study 1 and a control condition that wrote about a neutral stimulus.

## Study 1 method

### Participants

Thirty female undergraduate students 18 years of age or older at the University of Missouri participated in this study in exchange for partial course credit in a psychology course. We restricted our sample to females in Study 1 because we wanted to minimize the variability in narrative writing due to gender in order to maximize our effect size for the intervention. The mean age of the sample was 18.93 (SD = 1.10), and the majority of participants were White (83.3%). 6.7% of participants indicated that they were smokers, while 36.7% participants said one or both of their parents were smokers at any point in their childhood. 20% of participants indicated that their mother smoked.

### Materials and procedure

The study procedures and materials were reviewed and approved by the Institutional Review Board at the University of Missouri. The Institutional Review Board approved a waiver of written consent for this study. Participants instead read a cover letter describing the purpose of the study, the length of time the study would take, and additional information about privacy, confidentiality, and alternative assignments. Participants indicated consent by clicking a button at the bottom of the cover letter in the survey software that read, "If you choose to participate in this study, please click the arrow in the lower right-hand corner to begin."

**Narrative writing intervention.**   To construct a fictional narrative, participants were first asked to imagine a character that would be the protagonist in their story. In this exercise, they were asked to imagine a scene in which they are leaving a grocery store, and they witness a woman who is pregnant smoking a cigarette. After imagining this initial scene, participants were asked to respond to several items evaluating their attitudes toward the pregnant woman (measures described below). After the pre-writing assessment, participants were instructed to further develop the main character for their fictional writing task (e.g., the woman who smoked while pregnant) by imagining her age, race, economic status, educational status, etc.; see S1 Appendix for complete instructions for the narrative writing intervention. Participants also received instruction on creative writing more broadly and strategies for developing complex scenes with vivid detail and rich dialogue. The instructions were written by a member of the research team (L.S.) who has extensive experience with creative writing.

Participants then created two scenes in this fictional writing exercise that included their main character. In the first, they were instructed to draft a scene in which their character is on her way to work; in the second, they were asked to create a scene in which their character is attempting to do something difficult with another person. In addition, all participants were told to adhere to one rule while writing their narrative: they must assume that their character is at least as smart as they are. Smart was clarified to not mean "educated" but intelligent even

if a certain educational level had not been reached. Participants were reminded of this rule for both prompts. We included this stipulation because we wanted to go beyond the simple justification that the character made this choice (i.e., smoked while pregnant) because they did not understand the consequences. Rather, we wanted to nudge participants toward construction of some theory of mind that depicted their main character as capable of having a complicated inner life. Participants were also encouraged to write for a least five minutes for each of the two scenes. The next portion of the study was not accessible to participants until 5 minutes had passed.

## Measures

After imagining this scene but before engaging in the narrative writing task, participants were asked to rate their own emotional state on ten dimensions (sadness, excitement, anger, pity, disgust, happiness, hopelessness, surprise, disappointment, concern) from not at all (0) to very much (100). Following this, participants were asked about their attitudes toward the woman by rating their agreement from not at all (0) to very much (100) with the following statements: "The woman is a bad mother", "The woman is selfish", "The woman is doing the best she can" (reverse coded), "the woman does not have her future child's best interest at heart", and "I can never imagine a situation where I would smoke cigarettes while pregnant". To measure empathy and perspective taking, we adapted the Perspective Taking subscale of the Interpersonal Reactivity Index [43] so that the items reflected the main character in the narrative (e.g., "I put myself in her shoes") and used the same 0 (not at all) -100 (very much) scale to indicate their agreement with these statements. There were 14 total items measuring empathy and perspective taking. Participants were then presented with items intended to measure their attributions for the imagined woman's behavior, including: "To what extent is this person to blame for her action of smoking cigarettes while pregnant?", "To what extent are external factors, such as life circumstances, responsible for this person smoking cigarettes while pregnant?" and "To what extent does this person have freedom to make better choices?" After completing the narrative writing intervention, participants responded to the same items measuring emotions, attitudes, empathy/perspective taking, and attributions for the woman's behavior. The specific wording of all items can be found in S1 Appendix. All study materials, data, and R Markdown documents associated with this manuscript can be viewed at: https://osf.io/8fhrn/

Following the post-narrative assessment, participants were asked to complete a series of demographic questions.

We conducted a series of within-subjects analysis of variance (ANOVA) models comparing ratings before and after the narrative writing intervention to test our three hypotheses. Specifically, we predicted that the narrative writing exercise would increase empathy and perspective taking (H1), make attitudes towards the pregnant woman who smoked more positive (H2), and debias the fundamental attribution error (H3), whereby participants would be less likely to endorse internal attributions for the person's behavior.

To test H1, we reverse coded selected items so that higher values represented greater empathy and created a composite measure of empathy and perspective taking by summing the items we adapted from the Perspective Taking subscale of the Interpersonal Reactivity Index.

We had no *a priori* hypotheses about the effect of imagining a woman who smoked while pregnant on ratings of participants' current affective state. We conducted a series of within-subjects ANOVA models employing a Bonferroni correction for multiple comparisons to explore pre-post differences in ratings of mood state. The alpha level for these analyses was .005.

## Study 1 results

### Attitude, empathy/perspective taking, and attribution items

In support of H1, there were significant pre-post differences on empathy/perspective taking, $F(1, 29) = 54.46$, $p < .001$, $n^2_p = .65$. Mean scores at Time 1 were 531.2 (SD = 150.98), while mean scores at Time 2 were 790.80 (SD = 133.54), indicating that participants exhibited significantly more empathy and perspective taking after the narrative writing intervention than before.

In support of H2, there were significant pre-post differences on four of the five items measuring attitudes towards the character. Our planned analyses revealed that participants were less likely to rate the woman as: 1) a bad mother—M(SD)$_{pre}$ = 70.47 (24.14), M(SD)$_{post}$ = 50.46 (24.72); $F(1, 29) = 21.16$, $p < .001$, $n^2_p = .42$; 2) selfish—M(SD)$_{pre}$ = 77.83 (23.40), M(SD)$_{post}$ = 56.90 (25.53); $F(1, 29) = 15.28$, $p < .001$, $n^2_p = .35$; and 3) not having her child's best interests at heart—M(SD)$_{pre}$ = 83.90 (17.79), M(SD)$_{post}$ = 57.30 (25.44); $F(1, 29) = 30.34$, $p < .001$, $n^2_p = .51$. Participants were also more likely to rate her as "doing the best she could do" after the narrative writing intervention—M (SD)$_{pre}$ = 25.47 (20.98), M(SD)$_{post}$ = 60.47 (27.71); $F(1, 29) = 42.66$, $p < .001$, $n^2_p = .60$. However, there were no pre-post significant differences in participants' willingness to endorse the item, "I can never imagine a situation where I would smoke while pregnant"—M(SD)$_{pre}$ = 93.90 (13.37), M(SD)$_{post}$ = 90.87 (15.37); $F(1, 29) = 2.90$, $p = .10.$, $n^2_p = .09$.

Additionally, we found support for H3; we observed significant pre-post differences on all of the items measuring internal vs. external attribution. Our planned analyses showed that participants were: 1) less likely to believe that the woman is to blame for smoking cigarettes while pregnant—M(SD)$_{pre}$ = 91.77 (31.15), M(SD)$_{post}$ = 68.63 (26.22); $F(1, 29) = 27.23$, $p < .001$, $n^2_p = .48$; 2) less likely to believe that she has freedom to make better decisions—M(SD)$_{pre}$ = 95.17 (10.07), M(SD)$_{post}$ = 85.23 (18.43); $F(1, 29) = 10.99$, $p = .001$, $n^2_p = .27$; and 3) more likely to believe that external factors such as life circumstances are responsible for her behavior—M (SD)$_{pre}$ = 51.40 (30.57), M(SD)$_{post}$ = 72.73 (24.28); $F(1, 29) = 14.51$, $p < .001$, $n^2_p = .33$, after the narrative writing intervention as compared to before.

### Participant emotion

While employing the Bonferroni corrected alpha level (p < .005), participants still reported less negative affect after the narrative writing intervention than before. Specifically, participants felt less anger, $F(1, 29) = 27.52$, $p < .001$, $n^2_p = .49$, less disgust $F(1, 28) = 44.07$, $p < .001$, $n^2_p = .61$, less surprise, $F(1, 28) = 14.54$, $p = .001$, $n^2_p = .34$, and less disappointment, $F(1, 29) = 10.51$, $p = .003$, $n^2_p = .27$, as well as increased feelings of pity, $F(1, 29) = 17.56$, $p < .001$, $n^2_p = 0.38$. There were no significant pre-post differences in reported hopelessness, sadness, excitement, happiness, or concern, $p > .005$.

## Study 2 method

### Participants

One hundred and sixty-eight undergraduate students 18 years of age or older at the University of Missouri participated in this study in exchange for partial course credit in a psychology course. The mean age of the sample was 18.70 (SD = 0.95). The majority of participants were female (58.3%) and White (83.3%). 5.3% of participants indicated at they were smokers, while 63.2% participants said one or both of their parents were smokers at any point in their childhood. 15.5% of participants indicated that their mother smoked.

## Materials and procedure

The study procedures and materials were reviewed and approved by the Institutional Review Board at the University of Missouri. The Institutional Review Board approved a waiver of written consent for this study. Participants instead read a cover letter describing the purpose of the study, the length of time the study would take, and additional information about privacy, confidentiality, and alternative assignments. Participants indicated consent by clicking a button at the bottom of the cover letter in the survey software that read, "If you choose to participate in this study, please click the arrow in the lower right-hand corner to begin."

The design for Study 2 was the same as Study 1 with one exception. After imagining a scene in which they witness a pregnant woman smoking a cigarette and answering questions related to this scenario (e.g. current emotional state, attitudes towards the main character of the narrative, empathy/perspective taking, and internal vs. external attributions for the woman's behavior), participants were randomly assigned to either the narrative writing condition, which was comprised of the narrative intervention used in Study 1, or the control condition, in which participants were instructed to write about the room that they are sitting in. Like the narrative intervention, participants in the control condition were asked to create two scenes in their writing exercise. In the first scene, they were instructed to write about the room in which they were currently sitting; in the second, they were asked to create a scene that described what they saw as they first entered the building and walked to the room where the experiment was being held. Participants were also encouraged to write for a least five minutes for each of the two scenes. The next portion of the study was not accessible to participants until 5 minutes had passed.

As in Study 1, after completing either the narrative or control writing exercise, all participants were then asked to re-imagine the scenario in which they observe a woman a smoking a cigarette while pregnant, and to answer the same questions about their emotional state, their attitudes, empathy/ perspective taking, and their attributions for her behavior that they completed at the beginning of the study. These items were assessed at two time periods, before and after the writing exercises.

To examine the effectiveness of the narrative writing intervention in Study 2, we conducted a series of mixed-design ANOVAs on these measures examining interactions between time (pre-post difference) and condition (narrative intervention vs. control). We again hypothesized that the narrative writing intervention would increase empathy and perspective taking toward the main character of the narrative (i.e., woman who smoked while pregnant) (H1), result in a more positive attitude towards the character (H2), and induce greater external attribution for her behavior (H3), while the control condition would not change significantly on these measures. Hence, we predicted significant time x condition interactions on attitudes, empathy/perspective taking, and behavioral attributions.

We also measured participants' emotional responses to the scenario and conducted a series of mixed ANOVA models to look for time x condition interactions. As in Study 1, we had no *a priori* hypotheses about the effect of the interventions on the items measuring participant emotion. Therefore, we employed a Bonferroni correction to control for family-wise error with multiple comparisons. The critical value of alpha for these analyses was .005.

To test H1, we reverse coded selected items so that higher values represented greater empathy and created a composite measure of empathy and perspective taking by summing the items we adapted from the Perspective Taking subscale of the Interpersonal Reactivity Index.

In addition to comparing participant responses on the above items, we examined the content of the writing in both the narrative writing and control conditions using Linguistic Inquiry and Word Count, LIWC [44]. LIWC 2017 is a text analysis application designed to

examine the emotional, cognitive, and structural components of written and verbal speech. LIWC output includes word count, summary language variables (e.g., analytical thinking), general descriptors (e.g., words per sentence), linguistic dimensions (e.g., % of words in text that are pronouns), psychological constructs (e.g., affective processes), personal concern categories (e.g., work), and punctuation (e.g., periods). Our analyses will focus on the summary language variables and a subset of the psychological constructs including affective, social, and cognitive processes.

## Study 2 results

### Attitude, empathy/perspective taking, and attribution items

Replicating Study 1, we found support for all three hypotheses. We observed a significant time x condition interaction for the empathy/perspective taking items; we report the results from the individual items and the composite measure in Table 1. Change in empathy and perspective taking from before the writing exercise to after was significantly greater for participants in the narrative writing condition than participants in the control conditions (H1).

In support of H2, we observed significant time x condition interactions for all of the items measuring attitudes towards the woman who smoked while pregnant; see Table 1. For participants in the narrative writing condition, agreement with the statements "this woman is a bad mother", "she does not have her child's best interest at heart", and "this woman is selfish" decreased more after the intervention than participants in the control condition. Ratings of agreement with the statements "this woman is doing the best that she can" and "I can never imagine a situation where *I* would smoke cigarettes while pregnant" increased more after the intervention for participants in the narrative condition than for participants in the control condition.

In support of H3, there were significant time x condition interaction for two of these items measuring external attribution. Participants in the narrative condition were less likely to believe that the woman was to blame for her actions and more likely to believe that external factors were responsible for her behavior after their narrative writing exercise than before compared to participants in the control condition. There was no significant time x condition interaction for the item measuring her perceived freedom to make better decisions.

### Participant emotion

With the Bonferroni corrected alpha, we observed a time x condition interaction for anger—$F(2, 166) = 14.77$, $p < .001$, $n^2_p = .15$, disgust—$F(2, 166) = 5.92$, $p = .003$, $n^2_p = .07$, surprise—$F(2, 166) = 9.68$, $p < .001$, $n^2_p = .10$, disappointment—$F(2, 166) = 11.39$, $p < .001$, $n^2_p = .12$, and pity—$F(2, 166) = 12.15$, $p < .001$, $n^2_p = .13$. Participants in the narrative writing condition reported significantly less anger, disgust, surprise, and disappointment and more pity after the intervention than before compared to participants in the control group. There were no significant time by condition interactions observed for sadness, excitement, happiness, concern, and hopelessness, $p > .005$.

### LIWC analyses

There were a number of qualitative differences between the writing produced by the narrative and control conditions. First, participants in the narrative writing condition wrote almost two hundred more words on average than the control condition; see Table 2 for means, standard deviations, and statistics associated with all of the LIWC summary measures. Writing in the narrative condition was also characterized by significantly *less* analytical thinking, which

**Table 1. Attitude, empathy/perspective taking, and attribution items—Study 2, M (SD).**

| Perspective Taking Scale 0–100 Scale 'Strongly Disagree' to 'Strongly Agree' | Control | Narrative | Statistic | p | $n^2{}_p$ |
|---|---|---|---|---|---|
| Composite measure (sum of 14 items) | | | | | |
| Pre | 640.51 (161.60) | 646.65 (189.68) | F (2, 166) = 23.95 | < .001 | .22 |
| Post | 661.95 (161.26) | 752.01 (202.17) | | | |
| I put myself "in her shoes". | | | | | |
| Pre | 43.01 (3.16) | 46.45 (3.14) | F (2, 166) = 12.71 | < .001 | .13 |
| Post | 45.81 (2.80) | 59.62 (2.79) | | | |
| I felt very sorry for her when I was thinking about her problems. | | | | | |
| Pre | 41.44 (3.23) | 46.27 (3.21) | F (2, 166) = 6.10 | .002 | .07 |
| Post | 42.46 (3.05) | 56.77 (3.03) | | | |
| I tried to take her side of the problem. | | | | | |
| Pre | 32.02 (2.64) | 35.78 (2.62) | F (2, 166) = 22.26 | < .001 | .21 |
| Post | 40.72 (2.72) | 52.22 (2.71) | | | |
| I felt kind of protective towards her. | | | | | |
| Pre | 22.15 (2.32) | 26.38 (2.30) | F (2, 166) = 15.89 | < .001 | .16 |
| Post | 27.18 (2.72) | 38.50 (2.70) | | | |
| I imagined what it was like to be her. | | | | | |
| Pre | 46.74 (3.02) | 43.09 (3.00) | F (1, 166) = 20.56 | < .001 | .20 |
| Post | 44.79 (2.69) | 58.52 (2.67) | | | |
| Her misfortunes disturbed me a great deal | | | | | |
| Pre | 55.06 (3.01) | 51.15 (2.99) | F (2, 166) = 4.31 | .01 | .05 |
| Post | 54.88 (2.94) | 58.12 (2.92) | | | |
| I tried to look at her side of the situation in addition to my own. | | | | | |
| Pre | 47.38 (2.88) | 47.99 (2.89) | F (2, 166) = 7.62 | < .001 | .08 |
| Post | 46.92 (2.53) | 57.50 (2.51) | | | |
| I found it difficult to see things from her point of view. | | | | | |
| Pre | 63.06 (2.70) | 61.41 (2.69) | F (2, 166) = 17.53 | < .001 | .17 |
| Post | 61.28 (2.77) | 50.70 (2.76) | | | |
| I was sure I was right about her, so I didn't waste much time considering her side of the situation | | | | | |
| Pre | 47.12 (2.86) | 43.58 (2.85) | F (2, 166) = 1.96 | .15 | .02 |
| Post | 48.93 (2.85) | 37.77 (2.84) | | | |
| I tried to imagine how things look from her perspective. | | | | | |
| Pre | 44.81 (2.79) | 48.71 (2.77) | F (2, 166) = 12.42 | < .001 | .13 |
| Post | 47.64 (2.56) | 59.73 (2.55) | | | |
| Based on my feelings while considering this scene, I would describe myself as a pretty soft-hearted person | | | | | |
| Pre | 50.95 (2.62) | 53.04 (2.61) | F (2, 166) = 0.45 | .63 | .01 |
| Post | 51.02 (2.70) | 54.23 (2.68) | | | |
| Before criticizing her, I tried to imagine how I would feel if I were in her place. | | | | | |
| Pre | 41.79 (2.91) | 44.77 (2.89) | F (2, 166) = 12.08 | < .001 | .13 |
| Post | 47.40 (2.62) | 57.23 (2.60) | | | |
| I felt pity for her when I was thinking about her experience. | | | | | |
| Pre | 48.72 (3.13) | 47.86 (3.11) | F (2, 166) = 2.99 | .05 | .03 |
| Post | 48.75 (3.04) | 54.59 (3.03) | | | |
| I didn't spend lots of time trying to get her point of view. | | | | | |
| Pre | 56.26 (2.95) | 53.58 (2.93) | F (2, 166) = 5.40 | .005 | .06 |
| Post | 52.39 (2.72) | 44.72 (2.70) | | | |

(*Continued*)

**Table 1.** (Continued)

| Attitude Items 0–100 Scale 'Strongly Disagree' to 'Strongly Agree' | Control | Narrative | Statistic | p | $n^2_p$ |
|---|---|---|---|---|---|
| This woman is a bad mother. | | | | | |
| Pre<br>Post | 67.69 (2.75)<br>64.31 (2.74) | 67.84 (2.73)<br>57.13 (2.72) | F (2, 166) =<br>10.55 | <<br>.001 | .11 |
| This woman is selfish. | | | | | |
| Pre<br>Post | 75.69 (2.50)<br>72.17 (2.58) | 76.97 (2.49)<br>65.45 (2.57) | F (2, 166) =<br>11.55 | <<br>.001 | .12 |
| This woman is doing the best she can. | | | | | |
| Pre<br>Post | 19.79 (2.12)<br>23.57 (2.57) | 23.08 (2.20)<br>38.36 (2.56) | F (2, 166) =<br>21.07 | <<br>.001 | .20 |
| This woman does not have her future child's best interest at heart. | | | | | |
| Pre<br>Post | 83.97 (2.35)<br>74.72 (2.85) | 81.37 (2.34)<br>69.02 (2.83) | F (2, 166) =<br>14.28 | <<br>.001 | .15 |
| I can never imagine a situation where *I* would smoke cigarettes while pregnant. | | | | | |
| Pre<br>Post | 90.37 (1.82)<br>88.20 (2.50) | 94.01 (1.81)<br>87.86 (2.49) | F (2, 166) =<br>3.16 | .04 | .04 |
| **External Attribution Items 0–100 Scale 'Not at all' to 'A great deal'** | **Control** | **Narrative** | **Statistic** | **p** | **$n^2_p$** |
| To what extent is this person to blame for her action of smoking cigarettes while pregnant? | | | | | |
| Pre<br>Post | 86.06 (1.72)<br>83.32 (2.09) | 86.65 (1.71)<br>77.50 (2.08) | F (2, 166) =<br>12.76 | <<br>.001 | .13 |
| To what extent are external factors, such as life circumstances, responsible for this person smoking cigarettes while pregnant? | | | | | |
| Pre<br>Post | 56.42 (3.17)<br>58.74 (3.15) | 52.72 (3.15)<br>62.38 (3.13) | F (2, 166) =<br>6.33 | .002 | .07 |
| To what extent does this person have freedom to make better choices? | | | | | |
| Pre<br>Post | 86.44 (1.85)<br>85.41 (2.09) | 89.15 (1.84)<br>85.58 (2.08) | F (2, 166) =<br>2.35 | .10 | .03 |

measures the extent to which the writers' used words suggesting formal, logical, or hierarchical thinking patterns [45].

The texts written within the two conditions also differed significantly in their emotional tone. The algorithm for this summary variable was constructed such that larger numbers indicate a more positive emotional tone and smaller numbers indicate a more negative emotional tone [46]. Writing in the narrative condition was significantly more negative than writing in the control condition. And was also described by significantly greater clout, which reflects verbal representation of social status, confidence or leadership [47]. However, writing in the control group had greater authenticity, which measures the extent to which people are personal and vulnerable in their writing.

**Table 2. LIWC Summary variables.**

| | Control Writing Exercise | Narrative Writing Exercise | F (1, 170) | p | $n^2_p$ |
|---|---|---|---|---|---|
| **Word Count** | 431.23 (147.27) | 627.47 (269.39) | 35.13 | < .001 | .17 |
| **Analytical Thinking** | 79.26 (17.39) | 67.28 (18.78) | 18.87 | < .001 | .10 |
| **Clout** | 28.66 (11.86) | 78.11 (22.31) | 329.40 | < .001 | .66 |
| **Authenticity** | 88.14 (14.28) | 21.98 (29.04) | 369.50 | < .001 | .68 |
| **Emotional Tone** | 38.60 (17.38) | 27.35 (18.24) | 17.14 | < .001 | .09 |

**Table 3. LIWC Affective process measures.**

|  | Control Writing Exercise | Narrative Writing Exercise | $F$ (1, 170) | $p$ | $n^2_p$ |
|---|---|---|---|---|---|
| **Affect** | 2.07 (1.27) | 3.79 (1.48) | 66.61 | < .001 | .28 |
| **Positive emotions** | 1.35 (0.96) | 1.79 (0.97) | 8.27 | .005 | .05 |
| **Negative emotions** | 0.66 (0.58) | 1.97 (1.07) | 98.85 | < .001 | .37 |
| **Anxiety** | 0.08 (0.16) | 0.45 (0.47) | 48.70 | < .001 | .22 |
| **Anger** | 0.11 (0.22) | 0.42 (0.42) | 37.13 | < .001 | .18 |
| **Sadness** | 0.22 (0.27) | 0.46 (0.42) | 19.25 | < .001 | .10 |

Focusing on the LIWC psychological constructs, the narrative writing texts were more affect rich and slightly, but significantly, more positive than the control writing texts; see Table 3. However, the overall difference in affect between the two groups was largely driven by the sizable difference in negative emotion between the two texts, with the narrative texts exhibiting much greater amounts of negative emotion than the control texts. More specifically, the narrative writing had significantly more anxiety, anger, and sadness than the control writing.

The narrative writing texts also include more content related to social processes, $F(1,170) = 619.62$, $p < .001$, $n^2_p = .78$, than the control writing texts. Specifically, the narratives passages were much more likely to include discussion about family, $F(1,170) = 84.08$, $p < .001$, $n^2_p = .33$, and friends, $F(1,170) = 27.95$, $p < .001$, $n^2_p = .14$, than the control writing. The narrative writing texts also include slightly more content describing cognitive processes than the control writing, $F(1,170) = 4.63$, $p = .03$, $n^2_p = .03$, although this difference is much smaller than the other differences in affect and social processes.

## Discussion

Across two studies, we examined the impact of a creative narrative writing intervention on empathy/perspective taking, attitudes towards a target character, and attribution of blame (internal vs. external). Study 1 found that after writing the narrative intervention, participants held more positive attitudes towards the woman who smoked while she was pregnant. Specifically, they were less likely to endorse statements that the character was a bad mother, selfish, and did not having her child's best interest at heart. Participants were also more likely to agree that the character was "doing the best she could", and they increased the amount of reported empathy and perspective taking after the narrative writing intervention as compared to before. Finally, participants were less likely to endorse internal attributions for the negative health behavior (e.g. "the woman is to blame") after the narrative writing exercise and more likely to endorse external attributions for her behavior.

Study 2 was designed to replicate and extend the results of Study 1 by including a control condition that wrote about the room they were sitting in and the use of text analysis to describe the differences in writing content between the control and narrative conditions. We observed significant condition by time interactions, demonstrating greater changes before and after the writing intervention in the narrative group than the control group. The narrative writing group reported larger increases in empathy and perspective taking, more positive attitudes toward the target character, and a greater endorsement of external attributions for behavior.

We gained additional insight into the mechanism through which those changes occurred by conducting a text analysis of the writing produced by each group in Study 2. Writing in the narrative condition was characterized by less analytical thinking, a more negative emotional tone, and more content related to social processes. These findings are consistent with other differences observed between in studies employing LIWC. Writing that is narratively-styled

typically exhibits lower levels of analytical thinking, focusing on "the here-and-now" and other personal experiences [45]. In contrast, the increased negative emotional tone and social interactions observed in the narrative writing texts are likely due to task differences. In the narrative writing task, one of the two writing prompts asked participants to specifically write about their character engaging in a difficult situation, which likely contributed to a greater use of negatively valanced words. Additionally, the narratives included social interactions at a much greater rate than the texts written by the control group. Again, this is largely because the control group was more likely to refer to themselves and their experiences, while the narrative group wrote about a fictional character.

## Strengths and limitations

Perhaps one of the most important strengths of the present research is its potential to address disparities in the health care that marginalized and vulnerable populations may receive. The patients who are most at risk for engaging in negative health behaviors also tend to be less educated and less wealthy, thus complicating their access to care and potentially increasing the negative biases that their providers may hold, restricting the care that their providers may assign to them. However, when health professionals are able to empathize with their patients, experience similar situations as patients themselves, or are asked to imagine how their decisions may affect those patients, they tend to provide better care [22–24, 39]. Engaging in a creative narrative writing intervention such as one presented here may be a relatively quick, economically viable way to improve patient relations and care in these most vulnerable populations.

Yet, there are also some limitations of this research that reduce its immediate generalizability. Study 1 used only female undergraduate students as participants to decrease variability in written responses. However, women are known to be more empathic and demonstrate greater perspective taking than males [48]. To address this limitation, we replicated the results in a sample containing both male and female undergraduate students. However, the reliance on an undergraduate sample still limits our ability to generalize these results to our desired population of health professionals.

## Future research

The results of the present studies significantly contribute to our understanding of empathy-based interventions for health-related behaviors, yet this work also raises important questions. For example, whereas the present study asked participants to create a fictional story about an individual engaging in a negative health behavior, would it be as beneficial for health professionals to create stories about nonfictional characters, such as their patients? Or, alternatively, if health professionals were provided with short, nonfiction biographies of their own patients, would this impact the care that they provide to those patients? Indeed, a similar practice known as "My Life, My Story" was developed at the William S. Middleton Memorial Veterans Hospital in Madison, Wisconsin, and was implemented in 2013 [49]. Short,1,000-word biographies of patients' lives are created in a collaboration between a writer and the patient and are then added to the patient's medical records. Patients and health professionals have responded extremely positively to the program, believing that it helps foster better relationships between patients and their health care teams.

With any of these kinds of real-world applications, it is critical to know how–and in what ways, and for how long–the interventions may affect health professionals' behavior. For example, it would be useful to know whether the effects of such an intervention could transfer to other patients in similar health situations, or if the intervention would only work for the one

patient into whose shoes the provider will step (or whose biography they will read). And of course, research should examine whether engaging in this type of creative narrative writing intervention more generally (and positively) impacts long-term skills in perspective-taking, empathy, and attribution allocation such that a health professional might be able to transfer these changes to other patients, other distinct health behaviors, or to the varying circumstances in the lives of their patients.

Together, these two studies demonstrate that the creative narrative writing intervention is an effective perspective taking exercise that can engender more positive attitudes toward characters engaging in negative behaviors and reduce blame and internal attributions for the behavior. These results are consistent with past research demonstrating the effectiveness of interventions based on creative writing and other creative outlets on empathy and attitudes [29, 50, 51] as well as attributions [30], and underscore the importance of delineating the mechanisms that cause these shifts in thinking. Future research will focus on understanding whether this type of intervention can impact the care that patients receive, with members of vulnerable populations who engage in negative health behaviors perhaps benefitting most of all.

## Supporting information

**S1 Appendix. Study materials.**
(PDF)

## Author Contributions

**Conceptualization:** Victoria A. Shaffer, Jennifer Bohanek, Elizabeth S. Focella, Haley Horstman, Lise Saffran.

**Data curation:** Victoria A. Shaffer.

**Formal analysis:** Victoria A. Shaffer, Haley Horstman.

**Methodology:** Victoria A. Shaffer, Jennifer Bohanek, Elizabeth S. Focella, Haley Horstman.

**Project administration:** Victoria A. Shaffer.

**Resources:** Victoria A. Shaffer.

**Software:** Victoria A. Shaffer.

**Writing – original draft:** Victoria A. Shaffer, Jennifer Bohanek, Elizabeth S. Focella.

**Writing – review & editing:** Victoria A. Shaffer, Jennifer Bohanek, Lise Saffran.

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
