## [Decision Letter · Decision Letter 0]

7 Aug 2019

PONE-D-19-16962

Encouraging Perspective Taking: Using Narrative Writing to Induce Empathy for Others Engaging in Negative Health Behaviors

PLOS ONE

Dear Dr Shaffer,

Thank you for submitting your manuscript to PLOS ONE. After careful consideration, we feel that it has merit but does not fully meet PLOS ONE’s publication criteria as it currently stands. Therefore, we invite you to submit a revised version of the manuscript that addresses the points raised during the review process.

Two reviewers' comments on your work have now been received; you fill them attached below. As you will see, both reviewers are aware of the novel and timely nature of your work, and have generally positive comments about your manuscript. Please note that they also advise a couple of important changes before your paper can be published. Most importantly, one of the comments from Reviewer 1 requires restructuring the layout of the paper, to make it clear that your are reporting two different studies. Reviewer 2 also recommends to make more explicit the limitations of the studies. I support both recommendations. If you are prepared to undertake the revision required, I would be very happy to consider recommending the paper for an eventual publication.

Please bear in mind the following standard caveat if and when you revise the paper: Inviting resubmission does not entail that the next version, or any subsequent version, will be accepted for publication. Moreover, the clarifications that result from the revision may reveal new issues, hitherto unnoticed, that preclude publication.

We would appreciate receiving your revised manuscript by Sep 21 2019 11:59PM. To enhance the reproducibility of your results, we recommend that if applicable you deposit your laboratory protocols in protocols.io, where a protocol can be assigned its own identifier (DOI) such that it can be cited independently in the future. For instructions see: http://journals.plos.org/plosone/s/submission-guidelines#loc-laboratory-protocols

We look forward to receiving your revised manuscript.

Kind regards,

Paula Pérez-Sobrino, Ph.D.

Academic Editor

PLOS ONE

Journal Requirements:

Reviewers' comments:

Reviewer's Responses to Questions

**Comments to the Author**

1. Is the manuscript technically sound, and do the data support the conclusions?

Reviewer #1: Partly

Reviewer #2: Yes

2. Has the statistical analysis been performed appropriately and rigorously? 

Reviewer #1: I Don't Know

Reviewer #2: I Don't Know

3. Have the authors made all data underlying the findings in their manuscript fully available?

Reviewer #1: Yes

Reviewer #2: Yes

4. Is the manuscript presented in an intelligible fashion and written in standard English?

Reviewer #1: Yes

Reviewer #2: Yes

5. Review Comments to the Author

Reviewer #1: Dr. Frans Derksen

Researcher Primary Health Care/ Gender & Women's Health

Frans.Derksen@radboudumc.nl

Radboud university medical center

Dept. of Primary and Community Care

P.O. Box 9101

6500 HB Nijmegen

The Netherlands

Review manuscript: Encouraging Perspective Taking: Using Narrative Writing to Induce Empathy for Others Engaging in Negative Health Behaviors

Reviewer: Frans Derksen

Comments to the authors:

Dear authors,

Prior Comments: This is a very helpful paper on an important topic. Because too little research has been performed about the interaction of narrative writing with empathy and how this improves the efficiency in patient-GP communication.

Reading the manuscript I realized that it describes two studies, which causes, in my opinion, a hybrid situation (see my remarks). That’s why my remarks mostly are structurally characterized and less about the content of the manuscript.

Abstract: presents the central themes sufficiently but can be more clear when it is presented in a structure such as: goal, method, results, discussion, conclusion.

Introduction: describes important topics, however it is too much and too long. Many topics belong, in my opinion, to the discussion. For instance in a paragraph as “comparison with other research”. Furthermore dated references are used (1993, 1997).

At the end of the discussion a description of a precise gap in research is lacking and the goal of the study has been described ambiguously. In my opinion due to the hybrid situation as described above.

After the general introduction a new introduction starts of study 1, that is confusing.

Method: is a very long paragraph. In my opinion it is useful to summarize. Moreover sometimes the method is mixed up with items from the results paragraph.

Results: after the results the description of the second study starts and after this description a discussion starts. And the discussion lacks a clear arrangement (summary of the study, strengths and limitations, comparison with existing literature) .

Sorry, but at that moment my interest in the manuscript has been disappeared because of the overload of information.

I surely hope it can be published after major improvement. I strongly recommend to separate both studies.

Reviewer #2: • Overall Impression: This study has value for interventions that may help mitigate bias, promote perspective-taking and empathy, and foster considerations for the impact of social determinants of health on patient decision-making and health outcomes.

• Recommendations for writing style/grammar:

o It may be slightly awkward to start the first sentence of the abstract with “People,”. Perhaps something like, “It is very common for people, including health professionals….” OR “Societal expectations of self-care and responsible actions toward others may produce bias against those who engage in perceived self-harming behavior. This is especially true for health professionals, who have dedicated themselves to helping reduce the burden of illness and suffering.”

o In the last sentence of the abstract, the definition of the adjective “dubious” with regard to behavior does not seem to align with the authors’ intended use. Perhaps “irresponsible,” “self-harming” or “adverse” would be clearer to readers?

o Typo, top of page 16, “. There were no prepost differences in reported hopelessness, sadness, excitement, happiness, or concern, p > ,05.” – comma should be replaced with a period to read “p> 0.05.”

o Page 20, for the sentence, “There were no significant time by condition interactions observed for sadness, excitement, happiness, concern, and hopelessness, p > .005.” – did you mean to say “p> 0.05.” rather than 0.005? My understanding is that you were stating that the results in these domains were not significant (eg, p not equal to or less than 0.05), correct?

o Statements in the Results section, such as, “This is unsurprising given that one of the two writing prompts asked participants to specifically write about their character engaging in a difficult situation.” would likely be better placed into the Discussion Section. Results should be stated as such, with any interpretation thereof held until the discussion.

• Study Materials & Study Design

o In future iterations of the study, consider adding “Other/Non-binary” to gender options

o Consider collecting demographic data that includes intended career pathway (helping professions vs other) and past experiences with smoking that might bias a participant (eg, does the study participant has a family member who smokes, or is the participant a current or past smoker? These are variables that may influence perspective-taking.

• Discussion

o I did not see any discussion of limitations to the study. This should be added to the Discussion section.

Examples:

• Limiting participants to females (study 1) significantly limits generalizability of results, as females have been shown in numerous studies to statistically significantly test at higher empathy levels than males.

• Limiting the study to undergraduate students also limits generalizability to health professionals.

6. PLOS authors have the option to publish the peer review history of their article (what does this mean?). If published, this will include your full peer review and any attached files.

Reviewer #1: Yes: Frans Derksen

Reviewer #2: No

---

## [Author Response · Author response to Decision Letter 0]

20 Sep 2019

Paula Pérez-Sobrino, PhD

Dear Dr. Pérez-Sobrino,

Thank you for the feedback on the revised version of our manuscript. We used the comments from the two reviewers to further refine the paper. In this letter, I will detail the changes that we have made to the manuscript in response to these comments. As requested, we have uploaded both a copy of the revised manuscript with the changes tracked and an unmarked copy.

Editor Comments:

Comment: Please ensure that your manuscript meets PLOS ONE’s style requirements.

Response: We updated our manuscript and file names accordingly.

Comment: Please provide additional details regarding participant consent. In the ethics statement in the Methods and online submission information, please ensure that you have specified (1) whether consent was informed and (2) what type you obtained (for instance, written or verbal, and if verbal, how it was documented and witnessed). If your study included minors, state whether you obtained consent from parents or guardians. If the need for consent was waived by the ethics committee, please include this information.

Response: Neither study included minor participants. This is clarified in the Participants section of both studies (see pages 10 and 16). We provided additional details about the consent process for both studies in the Materials and Procedure subsections (see pages 10-11 and page 16). 

Reviewer 1:

Comment: Abstract: presents the central themes sufficiently but can be more clear when it is presented in a structure such as: goal, method, results, discussion, conclusion.

Response: It is our understanding that the PLOS One journal style requires unstructured abstracts, so we did not modify the abstract as suggested.

Comment: Introduction: describes important topics, however it is too much and too long. Many topics belong, in my opinion, to the discussion. For instance in a paragraph as “comparison with other research”. Furthermore dated references are used (1993, 1997).

Response: We streamlined the Introduction and reduced the text by approximately 1,000 words. However, we disagree that the date of publication should be a reason to exclude citations from the manuscript. While we cite a handful of classic references, the majority of the research cited in the manuscript are much more recent publications.

Comment: At the end of the discussion a description of a precise gap in research is lacking and the goal of the study has been described ambiguously. 

Response: We revised the Introduction to include a precise description of the gap in the literature and described the goals of the two studies more explicitly (pages 6-7)

Comment: After the general introduction a new introduction starts of study 1, that is confusing.

Response: We removed the separate introductions to the two studies.

Comment: Method: is a very long paragraph. In my opinion it is useful to summarize. Moreover sometimes the method is mixed up with items from the results paragraph.

Response: We improved the efficiency of our Method section by removing material that was redundant with the Introduction. However, we prefer to err on the side of greater transparency and retain many of the methodological details. We moved the details about our analytical methods from the Results section to the Method section.

Comment: Results: after the results the description of the second study starts and after this description a discussion starts. And the discussion lacks a clear arrangement (summary of the study, strengths and limitations, comparison with existing literature).

Response: We added a section on the Strengths and Limitations of the research and included headers to emphasize the structure of the Discussion section (pages 22-25)

Reviewer 2:

Comment: Overall Impression: This study has value for interventions that may help mitigate bias, promote perspective-taking and empathy, and foster considerations for the impact of social determinants of health on patient decision-making and health outcomes.

Response: Thank you!

Comment: It may be slightly awkward to start the first sentence of the abstract with “People,”. Perhaps something like, “It is very common for people, including health professionals….” OR “Societal expectations of self-care and responsible actions toward others may produce bias against those who engage in perceived self-harming behavior. This is especially true for health professionals, who have dedicated themselves to helping reduce the burden of illness and suffering.”

Response: We appreciate the suggestion and have modified the first two sentences of the Abstract to include your suggested wording.

Comment: In the last sentence of the abstract, the definition of the adjective “dubious” with regard to behavior does not seem to align with the authors’ intended use. Perhaps “irresponsible,” “self-harming” or “adverse” would be clearer to readers?

Response: We replaced ‘dubious’ with ‘irresponsible’.

Comment: Typo, top of page 16, “. There were no prepost differences in reported hopelessness, sadness, excitement, happiness, or concern, p > ,05.” – comma should be replaced with a period to read “p> 0.05.”

Response: We corrected the punctuation. 

Comment: Page 20, for the sentence, “There were no significant time by condition interactions observed for sadness, excitement, happiness, concern, and hopelessness, p > .005.” – did you mean to say “p> 0.05.” rather than 0.005? My understanding is that you were stating that the results in these domains were not significant (eg, p not equal to or less than 0.05), correct?

Response: Because we had no a priori predictions regarding emotions, we employed a Bonferroni correction for multiple comparisons, which resulted in a critical alpha value of .005. However, we reminded the reader about the use of the Bonferroni procedure in that paragraph to eliminate confusion.

Comment: Statements in the Results section, such as, “This is unsurprising given that one of the two writing prompts asked participants to specifically write about their character engaging in a difficult situation.” would likely be better placed into the Discussion Section. Results should be stated as such, with any interpretation thereof held until the discussion.

Response: As suggested, we moved the interpretive statements about the LIWC analyses to the Discussion section.

Comment: In future iterations of the study, consider adding “Other/Non-binary” to gender options

Response: Thank you for this suggestion. We will be sure to do so in our future work.

Comment: Consider collecting demographic data that includes intended career pathway (helping professions vs other) and past experiences with smoking that might bias a participant (eg, does the study participant has a family member who smokes, or is the participant a current or past smoker? These are variables that may influence perspective-taking.

Response: Thank you for the suggestions about future demographic information to include. We did actually measure a few of the suggested items, but forgot to include them in the Participant section. As indicated in our experimental materials available on OSF, we collected the following three items about smoking: 1) Do you smoke cigarettes? (y/n); 2) At any point in your childhood, did one or both of your parents smoke cigarettes? (y/n/don’t know); 3) At any point in your childhood did your mother smoke? (y/n/don’t know). All of the demographic items are now summarized in the respective participant sections for both studies.

Comment: I did not see any discussion of limitations to the study. This should be added to the Discussion section. Examples: 1) Limiting participants to females (study 1) significantly limits generalizability of results, as females have been shown in numerous studies to statistically significantly test at higher empathy levels than males. 2)Limiting the study to undergraduate students also limits generalizability to health professionals.

Response: We added a discussion about the strengths and limitations of the work on pages 23-24. 

Thank you for the opportunity to submit a revision of the manuscript. We appreciate the helpful comments, and we believe they have led to an improved version of the manuscript. We look forward to hearing your decision.

Sincerely,

Victoria A. Shaffer, PhD

Department of Psychological Sciences

University of Missouri

208 McAlester Hall

Columbia, MO 65211

Email: shafferv@missouri.edu

---

## [Decision Letter · Decision Letter 1]

4 Oct 2019

Encouraging Perspective Taking: Using Narrative Writing to Induce Empathy for Others Engaging in Negative Health Behaviors

PONE-D-19-16962R1

Dear Dr. Shaffer,

We are pleased to inform you that your manuscript has been judged scientifically suitable for publication and will be formally accepted for publication once it complies with all outstanding technical requirements.

With kind regards,

Paula Pérez-Sobrino, Ph.D.

Academic Editor

PLOS ONE

Additional Editor Comments (optional):

Dear authors, I am pleased to inform you that both reviewers agree that their suggestions and comments have been appropriately addressed, and that the manuscript is entirely improved. Many thanks for your hard work and the great spirit with which you have incorporated the suggestions. Congratulations!

Reviewers' comments:

Reviewer's Responses to Questions

**Comments to the Author**

1. If the authors have adequately addressed your comments raised in a previous round of review and you feel that this manuscript is now acceptable for publication, you may indicate that here to bypass the “Comments to the Author” section, enter your conflict of interest statement in the “Confidential to Editor” section, and submit your "Accept" recommendation.

Reviewer #1: All comments have been addressed

Reviewer #2: All comments have been addressed

2. Is the manuscript technically sound, and do the data support the conclusions?

Reviewer #1: Yes

Reviewer #2: Yes

3. Has the statistical analysis been performed appropriately and rigorously? 

Reviewer #1: I Don't Know

Reviewer #2: Yes

4. Have the authors made all data underlying the findings in their manuscript fully available?

Reviewer #1: Yes

Reviewer #2: Yes

5. Is the manuscript presented in an intelligible fashion and written in standard English?

Reviewer #1: Yes

Reviewer #2: Yes

6. Review Comments to the Author

Reviewer #1: Every part of the manuscript is corrected. The manuscript is entirely improved and allows much more insight into the subject.

Reviewer #2: The authors have incorporated recommended edits. The revised manuscript is substantially stronger, clearer, more transparently and thoughtfully discussed. Well done!!

7. PLOS authors have the option to publish the peer review history of their article (what does this mean?). If published, this will include your full peer review and any attached files.

Reviewer #1: Yes: Frans Derksen

Reviewer #2: Yes: Jennifer R. Hartmark-Hill, MD, FAAFP

---

## [Editor Report · Acceptance letter]

8 Oct 2019

PONE-D-19-16962R1 

Encouraging Perspective Taking: Using Narrative Writing to Induce Empathy for Others Engaging in Negative Health Behaviors 

Dear Dr. Shaffer:

I am pleased to inform you that your manuscript has been deemed suitable for publication in PLOS ONE. Congratulations! Your manuscript is now with our production department. 

With kind regards,

on behalf of

Dr. Paula Pérez-Sobrino 

Academic Editor

PLOS ONE